

# Global concentrations of microplastic in soils, a review.

Frederick Büks[1] and Martin Kaupenjohann[1]

[1]Chair of Soil Science, Dept. of Ecology, Technische Universität Berlin, 10587 Berlin, Germany

*Correspondence to:* Frederick Büks (frederick.bueks@tu-berlin.de)

**Abstract.** Worldwide, microplastic (MP) has been commonly recognized as a threat for soil ecosystems. Terrestrial soils are widely contaminated by MP due to the application of sewage sludge and wastewater, plastic mulching, littering, the input of tire wear from roads and atmospheric deposition. Within the last decade, an increasing number of individual studies focused on item counts and masses of MP in different global soil environments.

We reviewed these studies to achieve a representative picture of common degrees of contamination. The majority of the prospected agricultural and horticultural sites was exposed to sewage sludge and mulching film application and showed concentrations of <13000 items kg$^{-1}$ dry soil and 4.5 mg kg$^{-1}$ dry soil. Microplastic concentrations in soils in the vicinity to municipal areas were thereby 10 times larger compared to rural sites. The measurement of masses was generally underrepresented compared to item numbers, and mass data were often generated from microscopic analyses by use of shape-to-mass models instead of direct measurement. Extreme values such as on industrial sites exceed the common concentrations by 2 to 4 orders of magnitude, which might be attributed not only to the land use, but also to the applied methods of measurement. Campaigns with focus on other entry pathways like composts, road dust runoff and littering or land uses like grassland, forest, fallow and wilderness as well as industrial sites and landfills were underrepresented or nonexistent. Background loads such as atmospheric deposition were often not excluded from the measurements and, thus, the studies might overestimate the contribution of the analyzed entry pathway to the total load. Other studies focused on light density MP e.g. from mulching films and therefore underestimated the amount of soil MP.

Despite these limitations, the data give an impression on the spectrum of global MP concentrations and are a good basis for experiments examining the effects of MP on exposed soils. Based on the collected data, we identified problems of past studies and recommend that future experimentation take into account standardized methods of extraction and quantification, a proper characterization of the sampling sites and their history as well as the exploration of yet underrepresented entry pathways and land uses.



## 1 Introduction

The impact of microplastic (MP) on global ecosystems is widely accepted and discussed in
many comprehensive reviews (e.g. Lambert et al., 2014; Bläsing and Amelung, 2018; Ng
et al., 2018; Schell et al., 2020). The contamination of the environment with plastic waste
started to raise our awareness of this extraordinary stable material step by step when
seabirds were found in the beginning 1960s, perished with their guts full of plastic debris
(Thompson et al., 2009). Microplastic, then, has started to be recognized in the marine
environment during the 1970s, when e.g. Gregory et al. (1978) reported high counts of it
on New Zealand beaches. Thus, the early research on MP was mainly focused on marine
and limnic environments resulting today in a multitude of studies and early comprehensive
data on marine or inland waters compared to studies on terrestrial environments. At first,
soils were ignored.

Plastic most probably got access to the manifold soil environments, when petroleum-
based consumer products like fashion made of synthetic fibers entered markets in the
second half of the 20$^{th}$ century (Geyer et al., 2017). Today, the input pathways of MP to
agriculture, horticulture, orchards, grassland and forest soils comprise the application of
sewage sludge (and also most probably digestates and composts of it), waste waters,
composted and fermented organic waste products, the weathering and – in extreme cases
– plowing of mulching foils as well as irrigation with water from contaminated lakes or
rivers (Steinmetz et al., 2016; Bläsing and Amelung, 2018; Weithmann et al., 2018). Also
the littering, decay and comminution of plastic wastes (Huerta Lwanga et al., 2017), the
dispersion from inappropriately managed landfills by leaching (Praagh et al., 2018; He et
al., 2019) and the eolian transport of small-sized MP cause the input to different soil
ecosystems even in remote areas (Razaei et al., 2019). Littoral areas such as floodplains,
river banks, tidal flats and beaches additionally get regular input of MP by diffuse sources
through the aquatic environment (Barnes et al., 2009). Evidence suggests that regular
application of MP leads to significant accumulation in soils (Corradini et al., 2019; van den
Berg et al., 2020).

Dumped into the soil, MP is supposed to influence physical properties such as water
holding capacity (WHC), processes like soil aggregation, the performance and composition
of the soil microbial community, the soil fauna and the flora (de Souza Machado et al.,
2018; Lehmann et al., 2019; Rillig et al., 2019; Büks et al., 2020a; Fei et al., 2020).
Although the number of studies on MP in soil environments has been rising in the last
decade (Fig. 1), there is still little knowledge on the concentrations of MP and its relation to
adverse effects.

The broad collection of data on the range of MP concentration, type, shape and size in
global soils is fundamental for the appropriate design of studies on physiochemical and
biological effects of soil MP. Within the European Union, there is to date no legislation on
monitoring soil MP, which could provide these data. Ongoing research projects, which
focus on impact assessments, struggle with the lack of knowledge on concentrations and
properties of environmental MP, when experimental setups are applied.





To overcome this lack of information, soil MP concentrations were recently predicted by
input models. Nizzetto et al. (2016) estimated that due to the application of sewage sludge
the load of MP to European agricultural sites is 5.8 kg ha$^{-1}$ a$^{-1}$ (1.6 mg kg$^{-1}$ a$^{-1}$). Similar
calculations based on the production of sewage sludge in Germany or the threshold for the
application of sewage sludge deviate from this value only by one order of magnitude (Büks
et al., 2020b).

The aim of this review is to collect data about common soil MP concentrations, sizes,
shapes and types under the influence of different anthropogenic parameters. We focus on
terrestrial soils and excludes subhydric and semisubhydric sites such as river and lake
shores, beaches, tidal flats, mangroves and lagoons. Studies on such systems appeared
some years earlier than investigations on terrestrial sites, are more numerous, in parts
very comprehensive and provide, together with manifold local studies, a more equally
distributed global data then ever collected for soil MP (e.g. Lots et al., 2017; Eo et al.,
2018; Karthik et al., 2018; Scheurer and Bigalke, 2018; Yu et al. 2018).

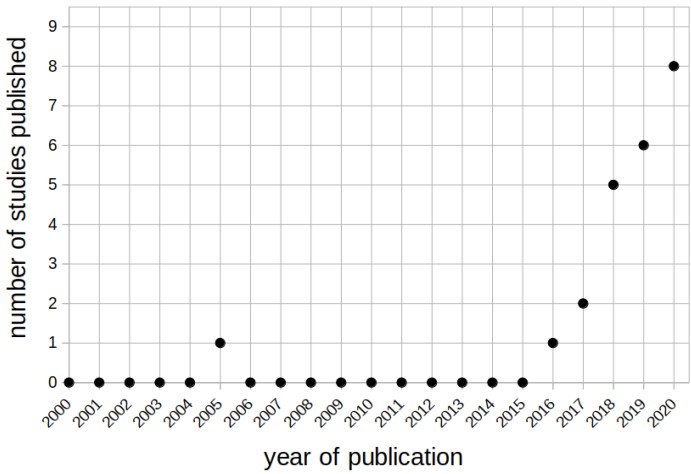

**Figure 1:** Number of studies on microplastic concentrations in terrestrial
soils published since year 2000.



## 2 Method


Our search was conducted by use of the Web of Science Core Collection Database and covers 23 studies with n=223 sampling sites published until August 2020 (Fig. 2). The search pattern contained the word "microplastic" linked in all possible combinations with each one of eight land-use types (the general term soil, agriculture, horticulture, orchard,

grassland, fallow, forest, landfill) and one of nine origins of MP (sewage sludge, waste water, plastic mulching, compost, digestates, road and tire wear, littering, flooding/ponding and eolian transport). The studies were searched for data on location, soil type, land use, origin of MP, vicinity (municipal, rural, industrial), sampling depth, method of MP extraction and measurement, type, size and shape of MP as well as MP concentrations based on

mass (mg kg$^{-1}$), items (items kg$^{-1}$) and particle surface (mm$^2$ kg$^{-1}$). All data refer to dry soil.

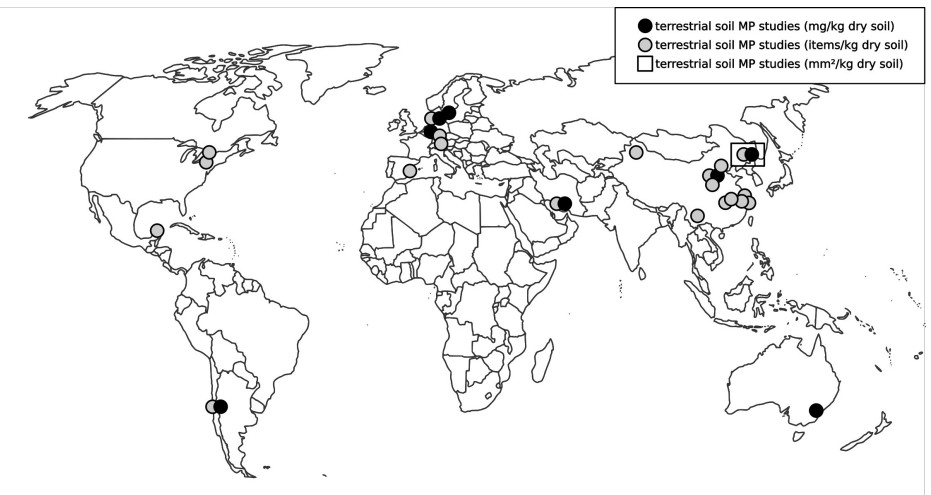

**Figure 2:** Global distribution of studies on soil microplastic concentrations (status August 2020).

Wherever possible, our approach uses one concentration value per each separated sampling site. Sites were considered as separated if they represent spatially separated landscapes, were differently managed or with clearly different soil properties. Some studies published multiple data sets on single sampling sites. In order to avoid statistical overvaluation of these sites, the data were pooled. Other studies, however, pooled values

from different localities to achieve more compact data. To compensate the loss of information, these data were regrouped based on the values provided in the published paper, the supplements or by use of raw data given by the authors. Data was also pooled if it was not clear weather it came from the same site. Data taken from different layers of the topsoil (e.g. 0-10 and 10-20 mm) were averaged to gain weighted mean

concentrations. In some cases, values could only be roughly extracted from figures, but with sufficient accuracy for the purpose of this work. If no raw data were provided by the authors and no structural adaptation of data was possible at all, the number of pooled sites



was noted and the average value was considered as one site. Detailed information can be found in the supplements.

Whenever median, minimum (min) and maximum (max) values where available, these data were favored over mean values ± standard deviation (sd). Stocks (per ha) were converted to concentration based of 1.2 g cm$^{-3}$ soil bulk density, regardless of the soil type. For the presentation of concentration ranges depending on the origin of MP, land use and vicinity, these data were grouped and the average values (medians and mean values) and

extrema (min and max values, mean values±sd) were plotted together with their collective median and interquartile range (25 % and 75 % quantiles). As these metrics are partly derived from pooled values, the calculated quantiles do not exactly represent the real quantiles. This is the reason, why a strong statistical analysis was not applied in this work.



### 3 Synopsis of regional microplastic concentrations

The 23 studies of this review were found to be very unevenly distributed around the globe. On Chinese territory, the only country in Eastern Asia that performed MP measurements, 11 regional studies with 155 sites were carried out mainly in the eastern coastal and the central region. On the European continent, six studies with 34 sites took place in Austria, Germany, Scandinavia and Spain, whereas no research was found on sites in further

European countries including Russia. North America has two studies with 8 sites, restricted to the northern regions. Only one study was carried out in Australia (1 site), the Middle East (10), South (5) and Central America (10), while Africa and the Indian subcontinent – certainly affected by the contamination of terrestrial soils with MP as well – conducted yet no investigation.


### 3.1 East Asia

In China, investigations were carried out with in part extensive number of sampling sites per study (Table 1). Most of the studies are structured very similarly using mechanical agitation within a salt solution for the detachment and density separation of MP. In two

cases a foregoing and in six cases a subsequent additional oxidation of soil organic matter with $H_2O_2$ or other oxidants was applied, followed by lightmicroscopy and FTIR spectroscopy (LM+FTIR) for particle counting and identification of MP (Liu et al., 2018; Zhang and Liu, 2018; Zhang et al., 2018; Han et al., 2019; Lv et al., 2019; Zhou et al., 2019; Chen et al., 2020; Ding et al., 2020; Huang et al., 2020; Zhang et al., 2020; Zhou et

al., 2020). The 11 studies showed a median MP particle number of 1076 items $kg^{-1}$ with a 25 % quantile of 78 items $kg^{-1}$, a 75 % quantile of 2500 items $kg^{-1}$ and a maximum of 690000 items $kg^{-1}$.

Three of these studies focused on mixed origins of soil MP and found average concentrations of 2625 items $kg^{-1}$ (n=38, 25 %: 1875 items $kg^{-1}$, 75 %: 14198 items $kg^{-1}$),

while no mass data was recorded (Zhang and Liu, 2018; Zhou et al., 2019; Ding et al., 2020). Agricultural land, that received sewage sludge application and plastic mulching, was examined by Ding et al. (2020) in vicinity to nine cities across the Shaanxi Province. The authors applied density fractionation (DF) with a cut-off of 1.5 g $cm^{-3}$ and a subsequent oxidation of soil organic matter (SOM). Subsequent LM+FTIR showed

concentrations of 2131±371 items of PE, PET, PP, PS and PVC per kg of dry soil mainly in the shape of fibers and fragments. Very similarly, Zhang and Liu (2018) sampled polymers from four agricultural sites with plastic greenhousing, sewage sludge and wastewater application and from one recently untreated afforested site near Kunming. Using oxidation of SOM and DF at 1.8 g $cm^{-3}$, they found average concentrations of 26070 items $kg^{-1}$

(min: 13470 items $kg^{-1}$, max: 42960 items $kg^{-1}$) in farmland Gleysols, 12050 items $kg^{-1}$ (min: 7100 items $kg^{-1}$, max: 26630 items $kg^{-1}$) in farmland Nitisols and 14440 items $kg^{-1}$ (min: 8180 items $kg^{-1}$, max: 18100 items $kg^{-1}$) in an afforested Gleysol, which indicates both the plastic load and the soil type to be factors of MP concentrations in soils. About 82% of the found items had sizes <250 μm, and fibers were the predominant shape

pointing out that this part came from wastewater origin. Much higher concentrations were



found by Zhou et al. (2019), who extracted plastic from urban fallows (min: 22000 items kg$^{-1}$, max: 200000 items kg$^{-1}$), horticultures (min: 43000 items kg$^{-1}$, max: 620000 items kg$^{-1}$) and forests (min: 96000 items kg$^{-1}$, max: 690000 items kg$^{-1}$) by use of a similar method with a cut-off at 1.55 g cm$^{-3}$. The samples largely originated from 180 sites near industrial areas in Wuhan.

Five studies with 90 sites examined the influence of plastic mulching and plastic greenhousing on MP concentrations on agricultural, horticultural and orchard plots (Liu et al., 2018; Zhang et al., 2018; Huang et al., 2020; Zhang et al., 2020; Zhou et al., 2020). In Shihezi, Huang et al. (2020) fractionated MP from cotton fields at 1.85 g cm$^{-1}$ with 185 subsequent H$_2$O$_2$ treatment. Concentrations measured by use of LM+FTIR increased with continuing application of plastic mulching from 80±49 (5 years) to 308±138 (15 years) and 1076±347 items kg$^{-1}$ (24 years). The exponential increase of particle number could be explained by intensified mulching or comminution of MP over time. By use of a similar method with a density cut-off at 1.6 g cm$^{-3}$, Zhou et al. (2020) quantified MP from 60 190 agricultural sites around the Hangzhou Bay. Average concentrations were 310 items kg$^{-1}$ (min: 0 items kg$^{-1}$, max: 2760 items kg$^{-1}$), and counts in areas with plastic mulching were more than twice as high as without. Unlike these two, Liu et al. (2018) used a lower density cut-off of 1.2 g cm$^{-3}$ to measure low-density MP in 20 plastic-mulched horticultures near Shanghai with 54.3 % of the items <1000 µm, mainly fiber-shaped, and concentrations of 195 only 70±13 items kg$^{-1}$. Due to the high number of sites with low concentrations pooled in Liu et al. (2018) and Zhou et al (2020), the actual average field concentration of MP is lower than calculated in this review.

The only East Asian studies that examined item and mass concentrations took place in rural sites on the Northern Chinese loess plateau (Zhang et al., 2018) and the city of 200 Harbin (Zhang et al., 2020). Both used distilled water for for the extraction of light-density MP. About 80±136, 87±213 and 187±222 items kg$^{-1}$ were measured on an agricultural field with plastic mulching as well as horticultural and orchard fields without plastic mulching (Zhang et al., 2018). Similar concentrations were found in rural and municipal areas near Harbin (Zhang et al., 2020). Both studies distinguished MP from other particles by use of 205 an innovative melting method and used a shape-to-mass model to derive mass concentrations from particle sizes (n=7, median: 0.1 mg kg$^{-1}$, 25 %: 0.0 mg kg$^{-1}$, 75 %: 0.4 mg kg$^{-1}$). Additionally, surfaces of 6.2±23.4 and 1.7±2.9 mm² kg$^{-1}$ were measured microscopically in municipal and rural samples, respectively (Zhang et al., 2020). The predominance of fibers in Liu et al. (2018) and Zhou et al. (2020) further suggest that past 210 or present application of sewage sludge or waste waters could not be ruled out for these sites.

Soils along busy roads are exposed to contamination with particles originating from road and tire wear. Around 1142 items kg$^{-1}$ (n=12, min: 300 items kg$^{-1}$, max: 12500 items kg$^{-1}$) were found in the vicinity of suburban roads in the municipal area of Wuhan Chen et al. 215 (2020). This is the only study on traffic-derived soil MP in China, and is only among two studies worldwide.





In a globally unique sampling, Lv et al. (2019) investigated ponding as an entry pathway of MP to rice cultures. The respective concentrations in paddy fields with and without rice-fish co-cultures, often used systems of production in Southeast Asia, were very low (up to 18±6 items kg$^{-1}$) between the rice-planting periods and more than twice as high in the planting season when the fields are ponded. Since the use of fresh water for irrigation represents a neglected entry pathway into agricultural soils, future investigation is needed, how soil MP concentration is affected by the amount of irrigated water, its MP load and the flooding regime.

Throughout the Chinese studies, four contained data on soils without statements on the origin of MP (NA). These samplings comprised very different sites such as the Nankai University campus in Tianjin (Han et al., 2019), horticultures in the vicinity of Wuhan municipal area (Chen et al., 2020), an afforested rural area near Kunming (Zhang and Liu, 2018) and both a rural horticulture and a orchard on the loess plateau (Zhang et al., 2018). These sites contained 118±190 items kg$^{-1}$ and 0.3±0.4 mg kg$^{-1}$ (n=3) in rural areas as well as 1142 items kg$^{-1}$ (n=10, 25 %: 938 items kg$^{-1}$ , 75 %: 4423 items kg$^{-1}$) in municipal areas. Moreover, there are no studies on areas with littering, the application of composts or digestates, only sewage sludge application or without any contamination.

**3.2 Europe**

Taking a look at the European continent, we see less studies with more heterogeneous loads and a focus on MP origins different from that in China. The average concentrations amount to 2914 items kg$^{-1}$ (n=30, 25 %=1332 items kg$^{-1}$, 75 %=8159 items kg$^{-1}$) and 8.9 mg kg$^{-1}$ (n=6, 25 %=0.3 mg kg$^{-1}$, 75 %=381.3 mg kg$^{-1}$) and are twice as high as in China.

Three studies with a total of 14 sites focus on the application of sewage sludge in farmland soils. They each used DF at 1.7 g cm$^{-3}$ followed by LM+FTIR analysis, but different pre-treatments with either $H_2O_2$ plus enzymes, detergents or no handlings (Vollertsen, 2017; Ljung et al., 2018; van den Berg et al., 2020). On croplands, that were largely located in rural areas of the Province of Valencia (Spain) and received an annual sludge application, the MP concentrations increased on average threefold within ten years of application (van den Berg et al., 2020), a rate that is in the range of Huang et al. (2020). The fields contained an average particle number of 3330 items kg$^{-1}$ (min: 999 items kg$^{-1}$, max: 8658 items kg$^{-1}$), whereas on untreated control plots only a third was counted. These values are 1 to 3 orders of magnitude lower than those found by Vollertsen (2017) in a Danish farmland soil, which contained on average 71000 items kg$^{-1}$ (mainly PE, PP and Nylon) on sites with sludge application and more than twice as much on the control site. Both studies, however, published concentrations equivalent to those in Chinese studies.

Vollertsen (2017) also estimated mass concentrations of 5.8 mg kg$^{-1}$ (min: 0.0mg kg$^{-1}$ , max: 16.5 mg kg$^{-1}$) after sludge application and 12.0 mg kg$^{-1}$ (min: 0.1 mg kg$^{-1}$, max: 224.3 mg kg$^{-1}$) on the control soil by use of a shape-to-mass model (Simon et al., 2018). This data largely correspond to other soils with low contamination. Due to issues



with the analysis, the author suggest that the data should be viewed as preliminary results. With a similar method Ljung et al. (2018) calculated mass concentrations from LM images

and identified MP concentrations of 0.3, 0.3 and 3.4 mg kg$^{-1}$ near Malmö (Sweden), that have been amended with singular 0, 1 and 3 t sewage sludge ha$^{-1}$ a$^{-1}$, respectively.

The only available European study on road dust was conducted by Dierkes et al. (2019). The authors used a pressurized liquid extraction (PLE) in combination with pyrolysis-gaschromatography-mass spectroscopy (Pyr-GC-MS) for the extraction and quantification

of PE, PP and PS from soil sampled adjacent to an arterial road near Köln (Germany). Mass concentrations found in this study (915±63 mg kg$^{-1}$) overshoot values measured in farmland soils by far and are more similar to industrial sides (Fuller and Gautam, 2016).

Without specification of entry pathways, extremely high particle counts were found in German and Austrian soils by LM after extraction with preliminary $H_2O_2$ treatment and

ultrasonication plus density fractionation (U+DF) at 1.45 g cm$^{-3}$ (Meixner et al., 2020). The method for manual LM counting included the support by an MP image reference database to identify items >5 µm. The extrapolated concentrations of 11x10$^6$ items kg$^{-1}$ (min: 2x10$^6$ items kg$^{-1}$, max: 26x10$^6$ items kg$^{-1}$) exceed values of other studies by 2 to 4 orders of magnitude.

The only study worldwide, that declared to work on an uncontaminated site, was conducted on a conventional farmland in Southern Germany with no plastic mulching or application of sewage sludge in advance (Piehl et al., 2018). The authors found very low concentrations of mainly films and fragments of PE, PP and PS. However, it is difficult to distinguish, whether the low counts of 0.31 items kg$^{-1}$ are caused by the low human input

or the method of extraction. Only particles >1000 µm were extracted by picking from $H_2O_2$ treated and sieved soil. Thus, smaller MP is ignored, which represent the majority of MP (Huerta Lwanga et al., 2017; Liu et al., 2018; Zhang and Liu, 2018; Razaei et al., 2019; Zhou et al., 2019; Chen et al., 2020; Ding et al., 2020). In addition, no further studies were conducted in Europe with focus on mixed contamination, plastic mulching, littering,

ponding or the application of composts or digestates. Nearly no data on the shape and size of collected MP were recorded.

### 3.3 The Americas

Three of four pan-american studies focus on sewage sludge application and found fibers

more than other shapes (Zubris and Richards, 2005; Corradini et al., 2019; Crossman et al., 2020). All sites contained on average 1190 items kg$^{-1}$ (n=37, 25 %=286 items kg$^{-1}$, 75 %=2060 items kg$^{-1}$) with the lowest concentrations in Ontario (Canada). Here, agricultural landscapes received repeated application of sludge and were examined by use of DF and LM+FTIR (Crossman et al., 2020). The MP concentrations increased from

4 items kg$^{-1}$ on a site with no application to 541±305 items kg$^{-1}$ after two applications. Zubris and Richhards (2005) extracted MP from four sites in the State of New York (USA) with DF in water and quantified only fibers by polarized LM. The authors found about 1235 items kg$^{-1}$ (min: 370 items kg$^{-1}$, max: 2060 items kg$^{-1}$), which is in the order of





magnitude like in global soils with sewage sludge application. A comparison with zones of
preferential flow shows an enhanced transport of MP fibers through macropores.

A comprehensive work conducted on agricultural land in the vicinity of Melipilla, Región
Metropolitana de Santiago (Chile) reported very similar concentrations (Corradini et al.,
2019). During 5 years, dewatered sewage sludge from a communal wastewater treatment
plant was 1-5 times applied with an annual amount of 40 t ha$^{-1}$, an usual amendment. After
the end of application in the year 2017, concentrations ranged from 1200 items kg$^{-1}$
(min: 0 items kg$^{-1}$, max: 2200 items kg$^{-1}$) in plots with one application to 3600 items kg$^{-1}$
(min: 1000 items kg$^{-1}$ to 10200 items kg$^{-1}$) after five applications. In addition, the authors
used a shape-to-mass model to estimate mass concentrations between 1.4 mg kg$^{-1}$ and
4.4 mg kg$^{-1}$.

In a worldwide unique study, Huerta Lwanga et al. (2017) quantified littered MP in rural
gardens on the Yucatan Peninsula (Mexico). The team extracted low-density MP by use of
U+DF in distilled water. About 95 % of the extracted plastic had a size <50 µm and  in total
amounts to 870±1900 items kg$^{-1}$. By use of a denser separation medium, the actual values
might have been higher due to the additional yield of denser plastics.


### 3.4 Middle East

In order to explore the translocation of soil MP by wind erosion, Razaei et al. (2019)
measured the concentration of low-density MP in soils of the (semi-)arid Fars Province
(Iran). Using a flotation method for the extraction of low-density MP in water (Zhang et al.,
2018) and LM, 1.2±0.6 mg kg$^{-1}$ and 205±186 mg kg$^{-1}$ were detected on five agricultural
sites and only 0.2±0.1 mg kg$^{-1}$ and 38±17 items kg$^{-1}$ on rangelands. Inadequate removal of
plastic mulch films was assumed to be the main origin of MP in these areas.

### 3.5 Australia

Fuller and Gautam (2016) used a pressurized fluid extraction (PFE) in combination with
gravimetric quantification and identification by FTIR to measure different plastics <1 mm in
soils near an industrial facility in Sydney (Australia). In accordance with the high potential
of contamination on industrial sites, which was also found by Zhou et al. (2019), the
recorded 2400 mg kg$^{-1}$ (min: 300 mg kg$^{-1}$, max: 67500 mg kg$^{-1}$) must be referred as "highly
contaminated".





**Table 1:** Studies on microplastic concentrations with characterization of sites, applied methods and extracted microplastic samples. The abbreviations used in this table are as follows: hort=horticulture, agr=agriculture, gra=grassland, orch=orchard, for=forest, fal=fallow, sew=sewage sludge application, pm=plastic mulching, pg=plastic greenhouses, WW=waste water, S=stirring, U=ultrasonication, AM=air-mixing, DF=density fractionation, SDS=Sodium Dodecyl Sulfate, LM=lightmicroscopy, FTIR=Fourier-transform infrared spectroscopy, Pyr-GC-MS=pyrolysis-gaschromatography-massspectrometry, Raman=Raman spectroscopy, flb=fibres, mb=microbeads, frag=fragments, pel=pellets. NA denotes that information was not available. Detailed data are listed in the supplements.

| region | sites | land use | entry pathway | vicinity | sampling depth (cm) | method of extraction | method of quantification/ qualification | measured plastic type (**bold**=found) | mg kg$^{-1}$ dw median mv ± sd (min–max) | items kg$^{-1}$ dw median mv ± sd (min–max) | size span [μm] | shape | reference |
|---|---|---|---|---|---|---|---|---|---|---|---|---|---|
| Wuhan, 武汉市 (China) | 8 | hort | NA | municipal | NA | S-DF (ZnCl$_2$ 1.5 g cm$^{-3}$) | LM, Raman | PA, PP > PS, PVC > PE | NA | 1083 (600–3167) | 20–5000 70%<200 | flb,mb>frag | Chen et al. (2020) |
| | 12 | | road | municipal | | | | | | 1142 (300–12500) | | | |
| Región Metropolitana de Santiago (Chile) | 1 | agr | sew (1x) | municipal | 0–25 | S-DF (water, NaCl 1.2 g cm$^{-3}$, ZnCl$_2$ 1.55 g cm$^{-3}$) | LM, shape-to-mass | NA | 1.4 (0.7–2.2) | 1200 (0–2200) | <5000 mainly <1000x50 | flb>others | Corradini et al. (2019) |
| | 1 | agr | sew (2x) | | | | | | 2.0 (1.8–3.2) | 1800 (1200–3200) | | | |
| | 1 | agr | sew (3x) | | | | | | 2.2 (0.6–4.6) | 1200 (200–4400) | | | |
| | 1 | agr | sew (4x) | | | | | | 2.9 (1.8–12.9) | 2200 (800–12400) | | | |
| | 1 | agr | sew (5x) | | | | | | 4.4 (1.8–10.3) | 3600 (1000–10200) | | | |
| Southeast Ontario (Canada) | 1 | agr | no sew | rural | 0–5, 5–10, 10–15 | Fenton's reagent, DF | LM, FTIR | PS, PE, PP, PU, polyester, others | NA | 4 ± NA | | flb>frag | Crossman et al. (2020) |
| | 2 | agr | sew (1x) | rural | | | | | | 103 ± 52 | | | |
| | 1 | agr | sew (2x) | rural | | | | | | 541 ± 305 | | | |
| Köln (Germany) | 1 | NA | road | municipal | NA | PFE | Pyr-GC-MS | PE > PP, PS | 915 ± 63 | NA | NA | NA | Dierkes et al. (2019) |
| Shǎnxī Province, 陝西省 (China) | 9 | agr | pm, sew | municipal | NA | S-DF (NaCl 1.2 g cm$^{-3}$, CaCl$_2$ 1.5 g cm$^{-3}$), H$_2$O$_2$ | LM, FTIR | PE, PET, PP, PS, PVC | NA | 2131 ± 371 | <5000 mainly <500 | flb>frag>other | Ding et al. (2020) |
| Sydney (Australia) | 1 | NA | NA | industrial | NA | PFE | gravimetric, FTIR | PE, PS, PVC, PP, PET | 2400 (300–67500) | NA | <1000 | NA | Fuller and Gautam (2016) |
| Tiānjīn, 天津市 (China) | 1 | NA | NA | municipal | NA | AM-DF (NaCl+NaI 1.5 g cm$^{-3}$), H$_2$O$_2$ | LM, FTIR | PP | NA | 95 | <3200 | frag | Han et al. (2019) |
| Shíhézǐ, 石河子市 (China) | 1 | agr | pm (5 yr) | municipal | 0–40 | S-DF (NaI 1.85 g cm$^{-3}$), H$_2$O$_2$ | LM, FTIR | PE | NA | 308 ± 138 | <5000 | frag | Huang et al. (2020) |
| | 1 | agr | pm (15 yr) pm (24 yr) | municipal | | | | | | 1076 ± 347 | | | |
| Yucatán Peninsula (Mexico) | 10 | hort | lit | rural | 0–10, 10–20 | U-DF (water) | LM | NA | NA | 870 ± 1900 | <2000 95%<50 | NA | Huerta Lwanga et al. (2017) |
| Shànghǎi, 上海市 (China) | 20 | hort | pg, pm | municipal | 0–3, 3–6 | U-DF (NaCl 1.2 g cm$^{-3}$), H$_2$O$_2$ | LM, FTIR | **PP, PE > polyester** | NA | 70 ± 13 | 20–10000 54%<1000 | flb>films>pel | Liu et al. (2018) |
| Malmö (Sweden) | 1 | agr | no sew | municipal | 0–20 | H$_2$O$_2$+enzymes, DF (ZnCl$_2$ 1.7 g cm$^{-3}$) | LM, FTIR, shape-to-mass | diverse | 0.3 | NA | 10–5000 | NA | Ljung et al. (2018) |
| | 1 | agr | sew (1 t ha$^{-1}$ a$^{-1}$) | municipal | | | | | 0.3 | | | | |
| | 1 | agr | sew (3 t ha$^{-1}$ a$^{-1}$) | municipal | | | | | 3.4 | | | | |
| Shànghǎi, 上海市 (China) | 3 | rice | ponding | municipal | 0–10 | S-DF (NaCl 1.24 g cm$^{-3}$), H$_2$O$_2$ | LM | NA | NA | 12 ± 4 | | flb, frag | Lv et al. (2019) |
| | 3 | rice-fish | | | | | | | | 4 ± 2 | | | |
| Austria/Southern Germany | 11 | NA | NA | municipal | NA | H$_2$O$_2$, U-DF (ZnCl$_2$ 1.45 g cm$^{-3}$) | LM | NA | NA | 11×10$^4$ (2×10$^4$–26×10$^4$) | 5–1000 | NA | Meixner et al. (2020) |





**Continuation of Table 1 ...**

| Location | n | land | treatment | | setting | depth | extraction | analysis | polymers | | conc (g) | size | shape | reference |
|---|---|---|---|---|---|---|---|---|---|---|---|---|---|---|
| Mittelfranken (Germany) | 1 | agr | none | | rural | 0-5 | $H_2O_2$, sieving | LM, FTIR | PE, PP, PS | NA | 0.3 (0.0-1.3) | 1000-5000 | films>frag>others | Piehl et al. (2018) |
| Fars province فارس (Iran) | 5 | agr | pm | | rural | 0-10 | S+U-DF (water) | LM, shape-to-mass | NA | 1.2 ± 0.6 | 205 ± 186 | 40-740 mainly <100 | NA | Razaei et al (2019) |
|  | 5 | gra | NA | | rural |  |  |  |  | 0.2 ± 0.1 | 38 ± 17 |  |  |  |
| Valéncia (Spain) | 1 | agr | sew (1x) | | rural | 0-10, 10-30 | S-DF (water, NaI 1.7 g cm³) | LM, FTIR | NA | NA | 1499 (999-1998) | >11 | frag>fib,films | van den Berg et al. (2020) |
|  | 1 | agr | sew (3x) | | rural |  |  |  |  |  | 2664 (999-3996) |  |  |  |
|  | 1 | agr | NA | | NA |  |  |  |  |  | 1998 (999-3663) |  |  |  |
|  | 1 | agr | sew (3x) | | rural |  |  |  |  |  | 2830 (1998-3330) |  |  |  |
|  | 1 | agr | sew (3x) | | rural |  |  |  |  |  | 5328 (1332-6327) |  |  |  |
|  | 1 | agr | sew (4x) | | rural |  |  |  |  |  | 3330 (1998-3996) |  |  |  |
|  | 1 | agr | NA | | NA |  |  |  |  |  | 7659 (7326-7992) |  |  |  |
|  | 1 | agr | sew (5x) | | rural |  |  |  |  |  | 3330 (1998-5328) |  |  |  |
|  | 1 | agr | sew (5x) | | NA |  |  |  |  |  | 2997 (2331-5994) |  |  |  |
|  | 1 | agr | sew (6x) | | rural |  |  |  |  |  | 3996 (1998-8658) |  |  |  |
|  | 1 | agr | sew (6x) | | rural |  |  |  |  |  | 2831 (1665-5994) |  |  |  |
|  | 1 | agr | sew (8x) | | NA |  |  |  |  |  | 2498 (333-4662) |  |  |  |
|  | 1 | agr | no sew | | municipal |  |  |  |  |  | 999 (333-2331) |  |  |  |
|  | 1 | agr | no sew | | NA |  |  |  |  |  | 500 (0-1332) |  |  |  |
|  | 1 | agr | no sew | | rural |  |  |  |  |  | 999 (0-1332) |  |  |  |
|  | 1 | orch | no sew | | rural |  |  |  |  |  | 999 (0-1332) |  |  |  |
|  | 1 | orch | no sew | | NA |  |  |  |  |  | 2664 (999-2664) |  |  |  |
| Denmark | 1 | agr | sew | | NA | 0-15 | SDS, S-DF (water, ZnCl₂ 1.7 g cm³) | LM, FTIR, shape-to-mass | **PP > PE >> nylon** | 5.8 (0.0-16.5) | 71000 (0-165000) | 20-500 | NA | Vollertsen (2017) |
|  | 1 | agr | no sew | | NA |  |  |  | **PE >> nylon > PP** | 12.0 (0.1-224) | 145000 (53000-528000) |  |  |  |
| Kunming, 昆明市 (China) | 2 (G) | agr | pg. sew, ww | | municipal | 0-5, 5-10 | NaOH+$H_2O_2$; U-DF (water, NaI 1.8 g cm³) | LM | NA | NA | 26070 (13470-42960) | 50-10000, 95% <1000, 82% <250 | fib>>others | Zhang and Liu (2018) |
|  | 2 (N) | agr | pg. sew, ww | | municipal |  |  |  |  |  | 12050 (7100-26630) |  |  |  |
|  | 1 (G) | for | NA | | NA |  |  |  |  |  | 14440 (8180-18100) |  |  |  |
| Loess Plateau, 黃土高原 (China) | 1 | orch | pm | | rural | 0-10, 10-30 | S+U-DF (water) | LM, shape-to-mass | **LD-PE, PP** | 0.3 ± 0.5 | 80 ± 136 | mainly >100 | NA | Zhang et al. (2018) |
|  | 1 | hort | NA | | NA |  |  |  |  | 0.5 ± 0.7 | 187 ± 222 |  |  |  |
|  |  |  | NA | | NA |  |  |  |  | 0.1 ± 0.1 | 87 ± 213 |  |  |  |
| Ha'erbin, 哈尔滨市 (China) | 2 | agr | pm | | municipal | 0-20 (0-30) | DF (water) | LM, FTIR, shape-to-mass | NA | 0.1 ± 0.6 | 163 ± 250 | 50-5000 mainly >100 | NA | Zhang et al. (2020) |
|  | 2 | agr | pm | | rural |  |  |  |  | 0.0 ± 0.0 | 75 ± 130 |  |  |  |
| Wuhan, 武汉市 (China) | 10 | hort | pm, sew, ww | | mainly industrial | at 5 | KOH+NaClO, S+U-DF (NaCl 1.19 g cm³, ZnCl 1.55 g cm³) | LM, Raman | **PE > PA, PP, PS** | NA | 43000-620000 | 10-5000 82% <100 | frag>others | Zhou et al. (2019) |
|  | 7 | for | pm, sew, ww | |  |  |  |  |  |  | 96000-690000 |  |  |  |
|  | 7 | fal | pm, sew, ww | |  |  |  |  |  |  | 22000-200000 |  |  |  |
| Hangzhou Bay, 杭州湾 (China) | 60 | agr | pm | | municipal | 0-10 | DF (NaCl 1.2 g cm³, NaI 1.6 g cm³), $H_2O_2$ | LM, FTIR | diverse | NA | 310 (0-2760) | >60 mainly 500-3000 | frag, fib>films | Zhou et al. (2020) |
| Ithaca (USA) | 1 | orch | sew | | municipal | 0-10, 10-25, 25-50 | S-DF (water) | LM | NA | NA | 1250 ± 60 | NA | fib | Zubris and Richards (2005) |
| Cobleskill (USA) | 3 | NA | sew | | municipal |  |  |  |  |  | 1240 ± 87 |  |  |  |



## 4 Global data assessment

### 4.1 Applied methods

The majority of studies in the present work recorded data in items kg⁻¹ (20 studies with 218 sites), eight studies contained data in mg kg⁻¹ on 29 sites and one study with 4 sites in mm² kg⁻¹. Five studies collected both number and mass concentrations on in total 24 sites, one applied all three measures.

The most commonly used method thereby comprised a treatment step to oxidize or detach soil organic matter in advance (in 7 cases) or subsequent (6) to a mechanical agitation followed by DF, LM counting and identification by use of FTIR (13) or Raman spectroscopy (2). The treatment was conducted in different combinations by use of $H_2O_2$ (10), Fenton's reaction, NaOH, KOH/NaClO, enzymes or detergents (each 1) and is lacking in eight 345 cases. Physical stressing of the soil matrix was performed by stirring/shaking eight times, ultrasonication (4), both (3) and bubbling (1), whereas any agitation lacks in five studies. Eleven studies used a density cut-off ≥1.5 g cm⁻³ to extract a collective of the most common types of plastic, whereas nine studies applied less dense extraction media. In conclusion, physical agitation and oxidation of organic matter together with fractionation in 350 a dense aqueous solution is the scheme most often used in advance to optical analysis of soil MP.

Six studies additionally applied a shape-to-mass model to estimate soil MP masses, and two studies measured MP masses directly from soil samples by use of PFE plus Pyr-GC-MS or PFE plus weighing. The TED-GC-MS, also listed in Bläsing and Amelung (2018) as 355 an adequate measurement method, has not been applied to quantify soil MP in the present field studies.

### 4.1 Global concentrations

Throughout all 223 examined sampling sites, the medians of soil MP are 1167 items kg⁻¹ 360 (n=218, 25 %: 89 items kg⁻¹, 75 %: 2870 items kg⁻¹) and 0.6 mg kg⁻¹ (n=29, 0.004 mg kg⁻¹, 2.65 mg kg⁻¹). Some studies, in parts conducted on industrial areas, exceed this values by orders of magnitude (Fuller and Gautam, 2016; Vollertsen, 2017; Dierkes et al., 2019; Zhou et al., 2019; Meixner et al., 2020). In the following, global data are pooled according to entry pathways, land use and vicinity (Figure 3).

The sites with sewage sludge application have average MP concentrations of 1998 items kg⁻¹ (n=24, 25 %: 999 items kg⁻¹, 75 % 3616 items kg⁻¹) or 2.2 mg kg⁻¹ (n=8, 1.7 mg kg⁻¹, 4.5 mg kg⁻¹), which increase with the number of sewage sludge applications (Corradini et al., 2019; Crossman et al., 2020; van den Berg et al., 2020). These loads are approximately one order of magnitude above values measured in fields with plastic 370 mulching, both in terms of number and weight. According to medians and quantile ranges of item numbers, sites with the input of road dust and littering are similar to sludge sites. However, just like paddy fields and uncontaminated plots, these data sets only base on




one study with a few sampling sites, and the data of mass concentrations are even sparser.

The number of MP items in fields with agriculture and horticulture/orchards is well investigated, which is in contrast to grassland, fallow and forest soils. Both have similar average concentrations of about 1200 items kg$^{-1}$ (n=118, 88 items kg$^{-1}$, 2830 items kg$^{-1}$) and 1000 items kg$^{-1}$ (n=57, 350 items kg$^{-1}$, 1604 items kg$^{-1}$), but in agricultural landscapes one encounters a much wider range of concentrations. A robust set of mass
concentrations is only found in agricultural soils and amounts to average concentrations of 1.7 mg kg$^{-1}$ (n=20, 0.3 mg kg$^{-1}$, 2.8 mg kg$^{-1}$). All other land uses have a minor number of studies and sampling sites.

Sites in the vicinity to municipal areas are three times more often investigated than rural areas. With 1275 items kg$^{-1}$ (n=147, 316 items kg$^{-1}$, 3005 items kg$^{-1}$) compared to
187 items kg$^{-1}$ (n=39, 0.3 items kg$^{-1}$, 1332 items kg$^{-1}$), the municipal measurements result in a particle number that is about one order of magnitude larger than in rural areas. This relation is also found in terms of MP masses that amount to 2.1 mg kg$^{-1}$ (n=11, 0.7 mg kg$^{-1}$, 4.5 mg kg$^{-1}$) and 0.2 mg kg$^{-1}$ (n=15, 0.0 mg kg$^{-1}$, 0.7 mg kg$^{-1}$), respectively. This implies a positive relation between population density and plastic concentration, which was also
shown along a river shore in Mongolia (Battulga et al., 2019). However, in some world regions, rural and urban areas overlap, especially in China, where suburban areas contain large agricultural sites and rural areas are often densely inhabited, leading to more similar concentrations in both vicinities. Measurements in highly contaminated industrial areas, although conducted only a few times, indicate concentrations 2 to 4 orders of magnitude
above non-industrial areas.

In conclusion, the most comprehensive data on both item number and mass are given for plastic mulching, sewage sludge application and multiple MP inputs as well as municipal and rural areas with food production. Based on overlapping interquartile ranges, the common concentrations in the respective categories range between <1 and
12760 items kg$^{-1}$ dry soil as well as 0 and 4.5 mg kg$^{-1}$ dry soil. In most cases extreme values do not scatter more widely than the medians, leading to the assumption that we are faced to a wide and even distribution of natural soil MP concentrations. Other data sets like on tire wear, forests, littering or industrial areas are sparse and might only give indications of expectable concentrations. In no region, studies on the application of MP
through composts and digestates were conducted. Also the contamination of wilderness soils was not in the focus of recent research on MP. In contrast, NA data were numerous in most categories due to a lack of comprehensive documentation.



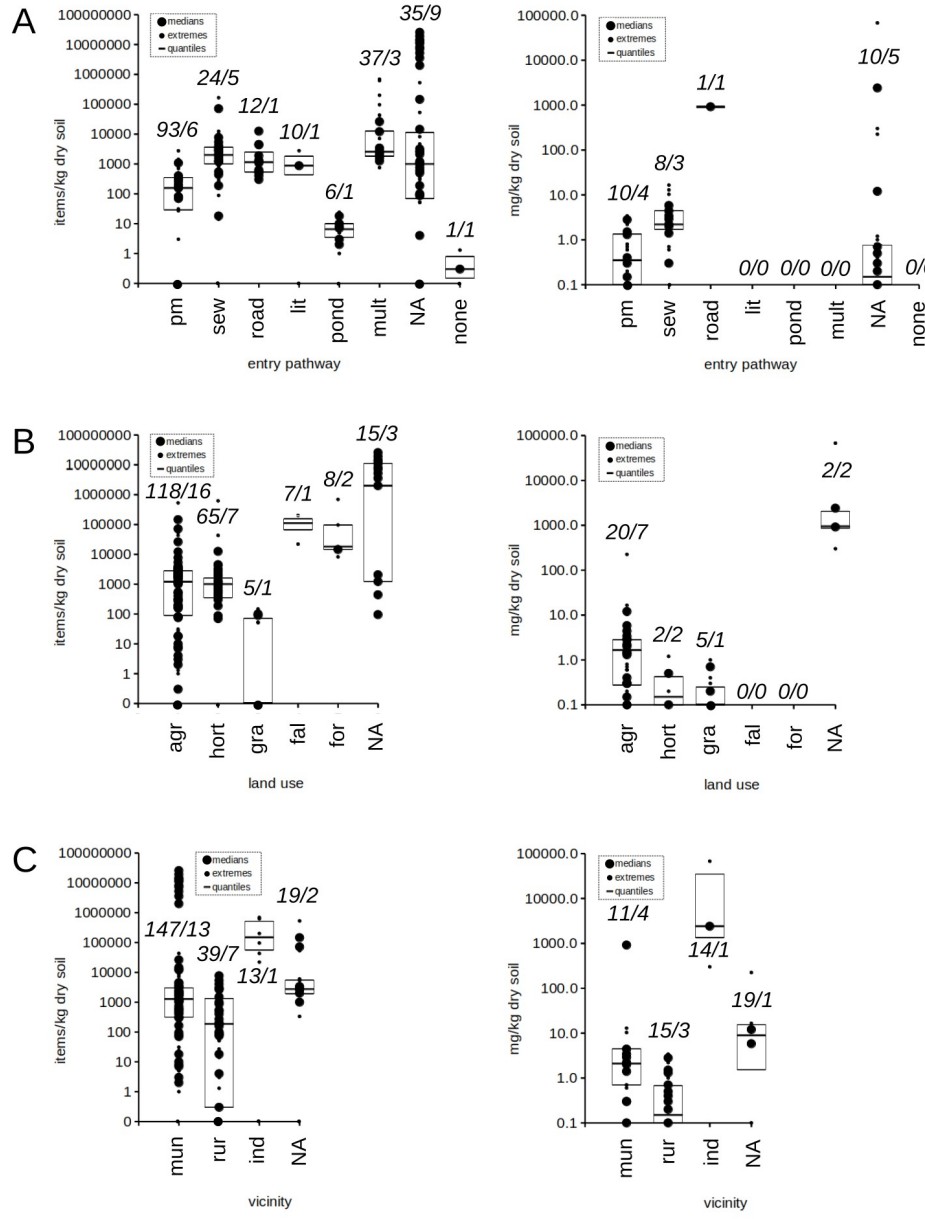

**Figure 3:** Concentrations of microplastic to soil ecosystems. Big dots mark median and mean concentrations measured in worldwide field experiments, small dots the related extreme values or standard deviations, strong horizontal lines the overall medians and narrow lines the 25 % and 75 % quantile. Italic numbers indicate the underlying number of sampling sites / studies. A: Entry pathways are pm=plastic mulching and plastic greenhousing, sew=sewage sludge and waste water application, road=road and tire wear, lit=littering, pond=ponding, mult=multiple, none=no existing entry pathways, B: Land uses are agr=agriculture, hort=horticulture and orchards, gra=grassland, fal=fallow, for=forest, C: Vicinities of the sampling sites are mun=municipal, rur=rural, ind=industrial. NA indicate data with no specification. No data were found on compost and digestate application, landfill soils or wilderness).





### 4.2 How robust are these data?

In this work 23 studies with in total 223 sampling sites were evaluated in terms of the entry pathways of MP to soils, the underlying land use and the vicinity of the investigated sites. Particularly in China, a large number of similarly structured studies provided a first spatial overview on common concentrations. Despite this satisfying yield of data our synopsis on MP concentrations in global soils must, however, be interpreted under certain restrictions.

On about 84% of the sites at least one MP entry pathway has been described and analyzed, whereas no information on origins (NA) was given on the remaining sites. However, further possible sources for MP inputs have neither been ruled out nor quantified in any study. For instance, the background pollution due to eolian deposition can be assumed to be ubiquitous even in remote global regions (Dris et al., 2016; Allen et al.,

2019; Evangeliou et al., 2020). Razaei et al. (2019) found 0.2±0.1 mg kg$^{-1}$ and 38±17 items kg$^{-1}$ related to air transport in sparsely populated grassland areas. Throughout the world, the deposed MP concentrations most likely decrease towards more remote areas, but are widely unknown. Along roads, there is also a contamination of soils with tire wear due to runoff, but also by windblown dispersal, that result in yet unknown spatial

concentration gradients (Dierkes et al. 2019). Last but not least, littering produces a series of locally randomized point loads, which increasingly appear in vicinity to urban areas, because most of the end-user waste is produced there. Those diffuse MP sources were hardly addressed within the reviewed papers and appear silently in addition to the entry pathways the studies focused on. Thus, their contribution to the total load is likewise an

unnamed part of the measured values. This results in an unknown overestimation of the loads coming on the focused entry pathways. Some of the studies additionally provide information that other entry pathways typically appear in their region, like irrigation with river water, use of mulch foils or the application of sewage sludge or waste water, but do not provide past and present information on their application on the measured plots. In

consequence, the classification of MP inputs can be improved by differentiated data on entry pathways and consequent inclusion of historical plot data.

Sparsely applied within the reviewed works, basic soil characterization is still an integral part of the description of sampling sites. Only 15 % of the sites are described sufficiently by means of soil texture or soil type in WRB or U.S. soil taxonomy. First data on Gleysols

and Nitisols, that imply that the soil type can influence the accumulation of MP in a way directly or indirectly (Zhang and Liu, 2018), show that standardized characterization of the soils will be helpful in future studies to relate the MP load with soil properties.

On most sites (81%), the sampling depths are well documented and vary between the top 5 cm, parts of the topsoil, the plow layer or deeper layers (Piehl et al., 2018; Zhang et al.,

2018; Huang et al., 2020). As the MP concentration is shown to vary with depth (Zubris and Richards, 2005; Liu et al., 2018; Zhang et al., 2018; Crossman et al., 2020), projecting concentrations of certain upper parts of a top layer to predict the total top layer's concentration will result in an overestimation of its MP stock. This might be less important in agricultural soils with annually plowed topsoils, but will affect the concentrations

estimated in soils with no tillage. In consequence, a documentation of the plowing regime



as well as a gradual and/or mixed sampling of the topsoil is strongly recommended. Furthermore, samplings in subsoils could give important information on the vertical translocation of MP towards the groundwater layer.

Organic matter is assumed to interfere with the measurements by binding plastic particles
to the mineral matrix during the extraction or being mistaken with plastic in the following microscopic examination. This could influence the yield of extraction and later identification of MP. For this reason, different pre- and post-treatments with $H_2O_2$ or other agents were applied in several studies. Reviewed data show that on average ~10x more particles were extracted by use of a pre-treatment compared to no or post-treatment (Figure 4). This
difference is also found in data generated by use of shape-to-mass models and not affected by the extreme high values found by Meixner et al. (2019).

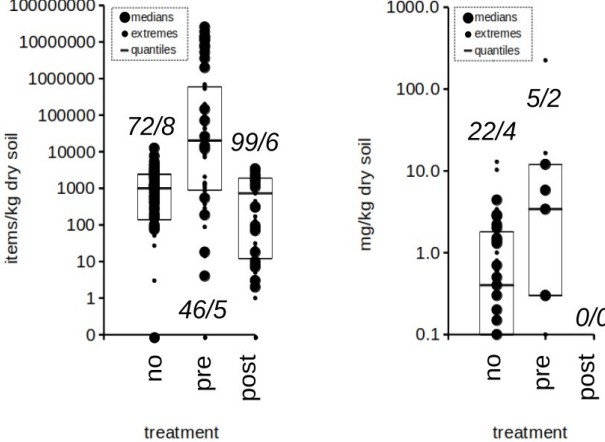

**Figure 4:** Quantities of MP measured after density fractionation without treatment of SOM (no), with pre-treatment (pre) and post-treatment (post). Big dots mark median and mean concentrations measured in worldwide field experiments, small dots the related extreme values or standard deviations, strong horizontal lines the overall medians and narrow lines the 25 % and 75 % quantile. Italic numbers indicate the underlying number of sampling sites / studies.


The total yield and composition of the extracted MP also strongly depends on the applied density cut-off. Some studies focused on a certain type of plastic such as light-density MP translocated by wind (Razaei et al., 2019) or fragmented PE foil, that was used for plastic mulching (Liu et al., 2018). Therefore, these studies used a less dense fractionation
medium (≤1.2 g cm$^{-3}$) and excluded denser MP from their measurements. However, only density cut-offs ≥1.5 g cm$^{-3}$ are suitable to extract nearly all types of plastic with relevant global production output (with the exception of chlorinated PVC and PTFE). On the other hand, densities ≥1.6 g cm$^{-3}$ cause the co-extraction of parts of the mineral matrix (Cerli et al., 2012). In consequence, the application of a density cut-off of 1.6 g cm$^{-3}$ like in soil



carbon pool analyses is recommended to avoid underestimation of the soil MP pool (Kaiser and Berhe, 2014). For such density cut-offs, sodium polytungstate (SPT) is an expansive but appropriate fractionation medium, as it can be adjusted to a wide range of densities (1.0 to 3.1 g cm$^{-3}$), is non-toxic, environmentally sound and recyclable (Six et al., 1999). Saturated NaCl solution, in contrast, only has a maximum density of 1.2 g cm$^{-3}$,
whereas $ZnCl_2$ and NaI are categorized as dangerous for the environment.

After density fractionation, the MP is often separated from the dense solution by use of a fine pored metal screen and cleaned by rinsing at the same place. To our knowledge, the smallest available mesh aperture is to date >5 μm (optimized dutch weave, GKD Group). Items with a profile smaller than this might become lost during the extraction. To date, we
do not know anything about the contribution of this fraction exactly due to this filtration problem. However, adverse effects on manifold parts of the soil fauna have shown to increase with decreasing MP particle size and irregular shape (Büks et al., 2020a; Büks et al., 2020b), which emphasizes the importance of small MP and nanoplastic analytics. In the reviewed studies, large percentages of items <250 μm were extracted independent of
the entry pathway (Table 1). The application of manual counting and the lack of defined lower size limits for counting cause imprecise quantification of this small MP fraction. Automated counting combined with particle sizing and shape analysis is therefore a coming challenge in MP analytics.

Not only the extraction but also the identification of MP types is crucial for the estimation of
their respective quantities in soil. From the eleven studies that used an appropriate density cut-off ≥1.5 g cm$^{-3}$, only two analyzed a broad set of common plastics. Two further studies used less dense extraction media or direct extraction by use of PFE in advance to analyze a similar spectrum of plastics, while all other studies tested ≤4 plastic types or lack any characterization in this category. For a better comparability of studies we suggest to
measure a broad set of commonly produced MPs by default plus other plastics with respect to the particular research.

The PFE method was only applied two times among the reviewed studies and showed values three orders of magnitude above all other mass values compiled by use of shape-to-mass models (Fuller and Gautam, 2016; Dierkes et al., 2019). It is unclear to what
extent this is caused by the selection of sampling sites, which are located within an industrial area and right next to an arterial road, or due to a significant mass fraction that has so far been ignored by use of optical methods but captured by PFE. At this stage, this cannot be estimated due to a small number of sites and the lack of comparative experiments between both the PFE and methods based on density fractionation.

Given data show, that sites with sewage sludge application have mass concentrations similar to those with plastic mulching, but ten times higher item concentrations (Corradini et al., 2019; Razaei et al., 2019). This points to smaller particle sizes in sewage sludge, which could be related to fibers (e.g. originating from textile cleaning), that appear more pronounced on sites with sewage sludge application (Table 1). That could mean that the
size characteristics of extracted particle collectives are strongly related to their entry pathways, so that we can estimate masses ex post from given data of item numbers? The





relationship of item and mass concentrations from data sets with both measures shows a different linear increase in soils with sewage sludge application ($R^2$=0.99, Corradini et al., 2019) and those with plastic mulching ($R^2$=0.67, Zhang et al., 2018; Razaei et al., 2019;

Zhang et al., 2020) (Figure 5). This pattern only bases on shape-to-mass data and is broken by the high concentrations found by Vollertsen (2017). However, within the concentration range found to be common in soils (<4.5 mg kg$^{-1}$ and <12760 items kg$^{-1}$) the two trends imply that the MP input from mulching films has a lower number of particles per mass than MP from sewage sludge. Derived from this data, the mass and number of

particles might roughly be estimated, if knowledge of the entry pathways exists. To support this statement, however, a larger amount of data is necessary.

In conclusion, most studies focus on item concentrations, some calculated masses by use of a shape-to-mass model and only a few measured MP masses directly. This results in less mass data and especially a lack of data generated by mass spectroscopic methods.

Number data, however, give a simple load indicator but no clear characterization about the soil MP load as the particle size distribution can vary strongly between different entry pathways and sites (Table 1). Some of these studies therefore conducted a rough size classification, that address ecological relevant properties such as bioavailability and perculation. In combination with a valid shape-to-mass-model, particle-sizing might be a

helpful tool for MP quantification in terms of both particle number and mass, especially in case of low financial and methodical capabilities. An identification of the given plastic types can be additionally performed by use of FTIR and Raman spectroscopy. The application of PFE, TED-GC/MS and Pyr-GC/MS, which is promising for precise mass analytics, in turn eliminates information about size and shape of the extracted plastic and therefore requires

additional particle-sizing.

A measure, which is only applied once in the reviewed studies, is the concentration of MP surfaces (mm$^2$ kg$^{-1}$ dry soil). To date, little is known about the relationship of the specific surface of MP items to adverse effects on soil organisms, transportation and occlusion within the soil matrix. In future experiments on soil contamination measurements of

specific MP surfaces in soils could play an important role. Furthermore, coming research on soil MP must not ignore plastic items >5 mm. Especially plastic mulching (and also littering) cause the deposition of an amount of larger fragments (Ramos et al., 2015; Huang et al., 2020). Even if we entirely stop the MP input to global soil ecosystems, its weathering and comminution would be an important reserve pool that might lead to further

increase of soil MP concentrations in future.





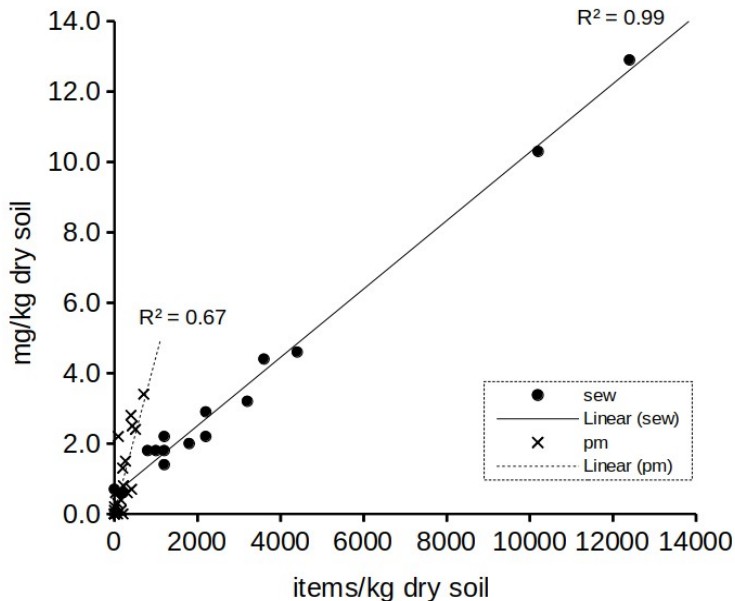

**Figure 5:** Regression of item and mass concentrations. Data were taken from sites with both measures (n=38). Dots represent measurements in soils with sewage sludge application, pm those with plastic mulching. Linear trands are shown by lines.





## 5 Conclusion

We reviewed 23 studies on soil MP concentrations with respect to the underlying entry pathways, land uses and vicinities. The in total 223 separated sampling sites were
predominantly located in China and Europe. The studies largely focus on sewage sludge application and plastic mulching on agricultural and horticultural sites near cities and on the countryside. In contrast, research on industrial and natural areas, the inputs of MP with road dust, littering, irrigation water, composts and digestates remain strongly underrepresented or lack in total. Common global MP concentrations amount of up to
13000 items kg$^{-1}$ dry soil and 4.5 mg kg$^{-1}$ dry soil, whereas concentrations on single plots exceeded these values. The mass and particle number introduced to global soils with sewage sludge is about one order of magnitude larger than with plastic mulching. According to other studies, the contamination in municipal vicinity is also one order of magnitude above that in rural areas, whereas concentrations in industrial areas overshoot
this values by far. We recommend to use common span to choose ranges of MP concentrations in coming laboratory experiment. Field studies are suggested to further record soil characteristics, plot history as well as entry pathways and conduct a gradual and/or mixed sampling of the topsoil and a sampling of the subsoil. If density fractionation is used to extract MP from soils, a 1.6 g cm$^{-3}$ dense medium is strongly recommended for
a sufficient separation of all common plastic types. A measurement of these types should be applied by default. Studies that determine particle numbers are invited to use a particle sizing with focus on the small-sized MP fractions, that were shown to have significant influence on the health of soil organisms. We hope that these points can contribute to establish a globally accepted pattern for soil microplastic measurements.





**Data availability**

All of the data are published within this paper and in the Supplement.

**Author contributions**

FB developed the review concept, collected data and prepared the paper. MK supervised
the study by participating in structural discussions on the idea and concept of the paper as
well as the final corrections.

**Competing interests**

The authors declare that they have no conflict of interest.


**Acknowledgements**

Best thanks to Esperanza Huerta-Lwanga, Fabio Corradini, Francisco Casado, Ines Fritz,
Jes Vollertsen, Katharina Meixner, Mona Kubiczek, Pim van den Berg, Pablo Meza, Raúl
Eguiluz and Violette Geissen for sharing their raw data.



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
