# Peer review of "Global concentrations of microplastic in soils, a review."

_SOIL, 2020_

## Referee Comment (RC1) · Anonymous Referee #1 · 27 Oct 2020

As microplastic has been commonly recognized as a threat for ecological environment and human health, and the research on microplastic pollution in terrestrial environment came into focus. The references on microplastic concentration in soils around the world was collected and analyzed in the present manuscript, this work is very important for the further research on microplastic in terrestrial ecosystem especially agricultural soil environment. Although the topic is of value for the scientific community, some questions should be considered before this paper can be considered for publication in Soil.

1. The methods for extracting microplastics from soils is not uniform among those studies listed in this reviewer, and different density suspensions can separated different types of microplastics. Whether is it meaningful to analyze and compare the concentration of microplastics extracted by different separation methods? E.g. line

272-274. . ...

2. The sample base is small, is the conclusion representative? E.g. line 20 "Microplastic concentrations in soils in the vicinity to municipal areas were thereby 10 times larger compared to rural sites"; line 368-370, the microplastic concentrations in sites with sewage sludge application are approximately one order of magnitude above values measured in fields with plastic mulching. . .. . .; line 385-386. . ..

3. Notices some of the details, please. E.g. line 119: 0-10 and 10-20 mm? line 200: double "for" etc.

---

## Referee Comment (RC2) · Anonymous Referee #2 · 4 Nov 2020

The authors discuss microplastic contamination and detail several limitations of methods currently in use. They rise several complementary research questions that should be tackled in future. Generally, the article is beautifully written. The idea of the article is original, figures and tables are clear. I recommend this article for publishing after some minor modifications: Line 85 Âń The aim of this review is to collect data about common soil MP concentrations, sizes, shapes and types under the influence of different anthropogenic parameters." I think the article goes further so I suggest improving this sentence. Based on the authors experience: could they add a table summarizing the pros and cons of each method for measuring MP in the soil. Line 365 -370 "...which increase with the number of sewage sludge applications (Corradini et al., 2019; Crossman et al., 2020; van den Berg et al., 2020)". What is the frequency of sewage adding?

[Figure]

Line 438 "..Only 15 % of the sites are described sufficiently by means of soil texture or soil type.." Was there any other information about soil characteristic beside texture, such as carbon concentration, macro-micro fauna activity, …? and how each parameter could be linked to MP concentration? If so it would be interesting to mention it if such data exists in the literature. I'm again sincerely grateful to the authors for providing such a high quality of research, the article was pleasant to read, well organized, and educational.

---

## Author Comment (AC1) · 5 Nov 2020

Dear referee #1

Thank you very much for your comments. In the following we want to explain how to consider your questions.

1. The methods for extracting microplastics from soils is not uniform among those studies listed in this reviewer, and different density suspensions can separated different types of microplastics. Whether is it meaningful to analyze and compare the concentration of microplastics extracted by different separation methods? E.g. line 272-274..... The density cut-off has indeed influence on the amount of extracted plastic as discussed in lines 466-480. However, we expect that this doesn't take place in a

degree which affects the order of magnitude (power of ten). Following PlasticsEurope (2016), the most commonly produced plastic types are PP ($\sim$0.91 g/cm$^3$), PE (0.87-0.97 g/cm$^3$), PVC (mostly 1.2-1.4 g/cm$^3$) , PU (1.0-1.25 g/cm$^3$), PET (1.38 g/cm$^3$) and PS (1.05 g/cm$^3$), and at different terrestrial sites, PE and PP, which are both low density plastics, were found to be much more abundant than PVC, PET and PU items (Büks et al., 2020a). As the great majority of studies used dense solutions with $\geq$1.2 g/cm$^3$, these light plastics are extracted unhindered by the solution. Future investigations with standardized methodology and a cut-off of 1.6 g/cm$^3$ will then give us data that include all plastics and are precise on a scale smaller than order of magnitude.

-> We propose to insert in line 470: "In different terrestrial environments, low-density plastics like PE and PP were found to be much more abundant than denser materials such as PU, PET and PVC (Büks et al., 2020a). The great majority of studies that used dense solutions with $\geq$1.2 g/cm$^3$ therefore extracted large parts of soil plastic independently from the chosen density cut-offs leading to trustable orders of magnitude."

2. The sample base is small, is the conclusion representative? E.g. line 20 "Microplastic concentrations in soils in the vicinity to municipal areas were thereby 10 times larger compared to rural sites"; line 368-370, the microplastic concentrations in sites with sewage sludge application are approximately one order of magnitude above values measured in fields with plastic mulching......; line 385-386....

-> Although this is a small number of separated sites compared to the possibilities of future comprehensive minitoring programs, the data shows clearly different clusters of concentrations and are therefore sufficient to give a first approximation of the order of magnitude of soil microplastic concentrations. For more exact determinations, indeed, additional measurements are needed especially in the underrepresented categories.

We therefore propose to insert in line 402: "Measurements with larger sets of separated sites e.g. in the frame of national monitoring programs will allow to estimate more precise and localized values of soil microplastic contamination."

3. Notices some of the details, please. E.g. line 119: 0-10 and 10-20 mm? line 200: double "for" etc.

->Thank you for your mindfull proofreading. In fact, these are cm. Done.

Best regards,

Dr. Frederick Büks and Prof. Dr. Martin Kaupenjohann

References

Büks, F., van Schaik, N. L., and Kaupenjohann, M.: What do we know about how the terrestrial multicellularsoil fauna reacts to microplastic?, SOIL, 6, 245–267, https://doi.org/10.5194/soil-6-245-2020, 2020a.

PlasticsEurope, Plastics - The Facts 2016: An Analysis of European Plastics Production, Demand and Waste Data (PlasticsEurope, 2016).

---

## Author Comment (AC2) · 5 Nov 2020

Dear referee #2

Thank you very much for that appreciating and motivating report.

Line 85: The aim of this review is to collect data about common soil MP concentrations, sizes, shapes and types under the influence of different anthropogenic parameters." I think the article goes further so I suggest improving this sentence.

-> We propose to add "... discuss the robustness of these data and give recommendations for future experiments."

Based on the authors experience: could they add a table summarizing the pros and

cons of each method for measuring MP in the soil.

->We add to line 540 "The compatibility of the underlying procedural steps with given requirements is listed in Table 2." and add a new table, which you can find attached.

Line 365-370 "...which increase with the number of sewage sludge applications (Corradini et al., 2019; Crossman et al., 2020; van den Berg et al., 2020)". What is the frequency of sewage adding?

-> The frequencies are now mentioned in line 295 "... after two applications within 5 years." (Crossman et al., 2020) and line 304 "... was 1-5 times applied with an annual rythm and amount of ..." (Corradini et al., 2019). The frequency in van den Berg et al. (2020) is already described in line 245.

Line 438 "Only 15 % of the sites are described sufficiently by means of soil texture or soil type.." Was there any other information about soil characteristic beside texture, such as carbon concentration, macro-micro fauna activity, . . .? and how each parameter could be linked to MP concentration? If so it would be interesting to mention it if such data exists in the literature.

->We would like to add to line 442: "Data on further parameters such as soil carbon content, micro- and macrofaunal activity, that are found to affect the aggregation, transport, comminution and decay of microplastics (Büks et al., 2020a), are largely missing."

Best regards,

Dr. Frederick Büks and Prof. Dr. Martin Kaupenjohann
* * *
**Table 2:** Evaluation of procedural steps used for the extraction of microplastics from soil. Extraction methods, additional treatments and methods for measurement appear in various combinations. Their compatibility with given requirements is shown based on the reviewed literature. Question marks indicate unknown performances, ρ refers to the density cut-off in g cm$^{-3}$, OM refers to organic matter, PFE refers to Presurized fluid extraction, FTIR refers to Fourier-transform infrared spectroscopy, Pyr-GC-MS refers to Pyrolysis–gas chromatography–mass spectrometry and TED-GC-MS refers to Thermal extraction and desorption–gas chromatography–mass spectrometry.

| | | increased yield of extraction | increased co-extraction of OM | co-extraction of mineral phase | determination of item number / size / shape | determination of MP mass | determination of plastic types | determination of MP surfaces |
|---|---|---|---|---|---|---|---|---|
| oxidation of natural OM | pre-oxidation | yes | ? | ? | yes | yes | yes | ? |
| | post-oxidation | no | | | yes | yes | yes | ? |
| extraction method | mechanical treatment | yes | yes | no | ? | yes | yes | ? |
| | density fractionation | ≥1.6 | ~ρ | >1.6 | yes | yes | yes | yes |
| | PFE | yes | ? | no | no | yes | yes | no |
| measurement | light microscopy | | | | yes | StM | no | no |
| | FTIR / Raman spectroscopy | | | | yes | StM | yes | no |
| | Pyr-GC-MS / TED-GC-MS | | | | no | yes | yes | no |

**Fig. 1.**